# Evaluating the Potential of LJ1-01 Nighttime Light Data for Modeling Socio-Economic Parameters

**DOI:** 10.3390/s19061465

**Published:** 2019-03-26

**Authors:** Guo Zhang, Xueyao Guo, Deren Li, Boyang Jiang

**Affiliations:** 1State Key Laboratory of Information Engineering in Surveying, Mapping and Remote Sensing, Wuhan University, Wuhan 430079, China; guozhang@whu.edu.cn (G.Z.); drli@whu.edu.cn (D.L.); 2Collaborative Innovation Center of Geospatial Technology, Wuhan University, Wuhan 430079, China; 3School of Remote Sensing and Information Engineering, Wuhan University, Wuhan 430079, China; jiangboyang@whu.edu.cn

**Keywords:** LJ1-01, NPP/VIIRS, socio-economic parameters, linear regression

## Abstract

The LJ1-01 satellite is the first dedicated nighttime light remote sensing satellite in the world and offers a higher spatial resolution than the Defense Meteorological Satellite Program’s Operational Linescan System (DMSP/OLS) and the Visible Infrared Imaging Radiometer Suite (VIIRS) sensor on the Suomi National Polar-orbiting Partnership (NPP) satellites of the United States. This study compared the LJ1-01 nighttime light data with NPP/VIIRS data in the context of modeling socio-economic parameters. In the eastern and central regions of China, 10 parameters from the four aspects of gross regional product (annual average population, electricity consumption, and area of land in use) were selected to build linear regression models. The results showed that the LJ1-01 nighttime light data offered better potential for modeling socio-economic parameters than the equivalent NPP/VIIRS data; the former can be an effective tool for establishing models for socio-economic parameters. There were significant positive correlations between the two types of nighttime light data and the 10 socio-economic parameters; that for the gross regional product was the highest.

## 1. Introduction

Socio-economic parameters are of great value in the context of government policies and scientific research. Due to the limitations of traditional statistical survey methods, the acquisition of socio-economic parameters often suffers from shortcomings, such as large errors and lack of spatial information [1]. Previous studies have shown that nighttime light remote sensing images are highly correlated with human activities [2,3,4,5] and offer the advantages of spatiotemporal continuity, independence, and objectivity. Elvidge et al. [6,7] demonstrated that the remote sensing of nocturnal lighting provides an accurate, economical, and straightforward way to map the global distribution and density of developed areas. This paper’s theoretical basis relies on such previous work. Elvidge et al. [8,9,10,11] evaluated the quantitative relationship between nighttime light images and the population and gross domestic product (GDP). Ma et al. [12] examined the high correlation between the population, GDP, electricity consumption, and area of paved roads with nighttime light data on a small scale. Therefore, nighttime light data can provide an important basis for the estimation of socio-economic parameters, such as the gross regional product (GRP) [13,14,15,16,17], annual average population [18,19,20], electricity consumption [2,3], and area of land in use [12,21].

In October 2011, the Visible Infrared Imaging Radiometer Suite (VIIRS) sensor with a national polar-orbiting satellite (Suomi-National Polar Orbiting Partnership, Suomi-NPP) was launched successfully. The VIIRS imaging width is 3000 km, with a spatial resolution of about 740 m. Compared to the traditional nighttime light remote sensing data offered by the Defense Meteorological Satellite Program’s Operational Linescan System (DMSP), Suomi-NPP is more sensitive to night light [8]. Elvidge et al. [22] detailed the improvements of the VIIRS relative to the DMSP. The low light imaging capabilities of the DMSP and VIIRS were originally intended to provide meteorologists with global nighttime visible cloud imagery using moonlight reflectance. The original design was not intended to detect electric lighting; this capability was only discovered when early images were viewed.

The LJ1-01 satellite is the first dedicated nighttime light remote sensing satellite in the world and was successfully launched on 2 June 2018. Its development was led by a scientific research team from Wuhan University, who focused on the need to test nighttime light remote sensing and navigation enhancement technologies in China. The satellite was manufactured by the Chang Guang Satellite Technology Co. LTD. Its spatial resolution is 130 m, higher than that of the DMSP/OLS and NPP/VIIRS satellites of the United States. As its range spans 250 × 250 km, global nighttime light image acquisition can be completed within 15 days under ideal conditions, representing a significant technological advance in Chinese remote sensing capabilities from surface monitoring to social and economic development monitoring. The LJ1-01 satellite provides thematic products, such as the GDP index, carbon emission index, urban housing vacancy rate index, and so on. It dynamically monitors macroeconomic operation in the world and provides an objective basis for government decision-making.

Therefore, the LJ1-01 data with high spatial resolution can provide a more accurate nighttime light source for economic modeling. Domestic LJ1-01 nighttime light data can be of great significance and value when applied to analyses of urban development and studies of social and economic environments. This study compared LJ1-01 and NPP/VIIRS nighttime light data and evaluated the potential of the former to model socio-economic parameters, such as night commercial activities and electricity consumption.

## 2. Materials and Methods

### 2.1. Study Area

In order to reflect the social and economic development of different regions in China accurately, the country was divided into four economic regions: East, middle, west, and northeast (Figure 1). The eastern region, which leads in economic development and has the most developed economy, includes Beijing, Tianjin, Shanghai, and the provinces of Hebei, Shandong, Jiangsu, Zhejiang, Fujian, Guangdong, and Hainan. The economy of the central region has grown the fastest; this includes the provinces of Shanxi, Anhui, Jiangxi, Henan, Hubei, and Hunan. This study selected 87 and 80 prefecture-level cities in the eastern and central regions, respectively, as the research focus; these are referred to hereafter as administrative units (Table 1).

### 2.2. Data Source

All LJ1-01 cloud-free nighttime light imagery data satellite data are provided to users free of charge [23]. Data are immediately available for teaching and scientific research for registered institutions, while commercial purposes require prior communicate and cooperation with the Hubei Province High-resolution Earth Observation System Data and Application Network. Users need to log in at http://www.hbeos.org.cn/, click “Data Application” at the bottom right of the home page to enter the “High Score Management Platform”, and click “LJ1-01” in the menu bar to enter the data download registration page. Users can navigate to the registration page or directly visit http://59.175.109.173:8888/app/login_en.html. Users can download data after registration.

In this study, the nighttime light remote sensing data product of the LJ1-01 satellite from December 2018 were used, as shown in Figure 2a. A representative image portraying the NPP/VIIRS nighttime light data as a synthesis of the cloud-free average monthly product [24] of April 2018 is shown in Figure 2b. NPP/VIIRS nighttime light data were obtained from the National Geophysical Data Center (NGDC) under the National Oceanic and Atmospheric Administration (NOAA) (http://ngdc.noaa.gov/eog/viirs/download_viirs_ntl.html).

The socio-economic statistics of prefectural units in China were obtained from the China City Statistical Yearbook. Unfortunately, the 2018 social and economic statistics have not yet been published, so those for 2016 (in the latest China City Statistical Yearbook, 2017 [25]) were adopted instead. In order to more fully contrast LJ1-01 and NPP/VIIRS nighttime light data with socio-economic parameters, 167 statistical units for 2016 were collected. A total of 10 types of socio-economic parameters were defined, related to the GRP, annual average population, electricity consumption, and area of land used; the specific statistical data classifications are shown in Table 2. The vector boundary data used to define the provincial and prefecture-level city administrative boundaries were obtained from the National Geomatics Center of China.

### 2.3. Data Preprocessing

Since the NPP-VIIRS data were not filtered to remove the light detection associated with volcanic activity and background noise, the data noise limited the reliability of the experiment, so preprocessing was carried out to reduce this effect [2]. The signal/noise ratio (SNR) of the LJ1-01 data is higher than 35 dB [26], and Nightsat lowlight imaging data must achieve minimal detectable radiances with an SNR of 10 dB [6]. Thus, it was not necessary to remove noise from the LJ1-01 data, which itself has already had negative pixels removed, so the noise of the NPP-VIIRS data was removed with LJ1-01 data as the template. Based on the assumption that both LJ1-01 and NPP-VIIRS data in 2018 had the same light-producing regions, the digital number (DN) value was extracted as a positive pixel from the 2108 NPP-VIIRS data, which were then superimposed on the 2018 LJ1-01 data [3]. Only pixels corresponding to the same position were retained, and a pixel with a negative DN value was assigned a value of 0 to generate the initially corrected image [2].

NPP/VIIRS data were corrected for absolute radiation, and provide radiation brightness values in. Therefore, absolute radiation correction was necessary for LJ1-01 data. The following radiance conversion formula was used:
(1)$$L={\mathrm{DN}}^{3/2}\cdot 10^{-10}$$
where L is the radiance value after absolute radiation correction in $W/\left(m^{2}\cdot \mathrm{sr}\cdot \mu m\right)$. The DN value is the image grey value for each pixel. The radiance of LJ1-01 was converted to the central wavelength, while the NPP satellite uses the full-band radiance. Therefore, the difference in bandwidth between the LJ1-01 and NPP data results in the difference in the unit of radiance value.

The vector boundary data for each prefecture-level city were used as the mask to extract the nighttime light images required by the experiments from the global data set of LJ1-01 and NPP/VIIRS nighttime light images. All data adopted the Lambert azimuthal equal area projection.

### 2.4. Linear Regression Model

Usually, the potential of the nighttime light data for modeling socio-economic parameters is evaluated by three regression analysis methods: The linear regression model, the logarithmic model, and the quadratic regression model. Of these, the linear regression model is relatively accurate and easy to realize and was therefore adopted in this study as follows:(2)G=a·TNL+b
where G is the sum of socio-economic parameters in the administrative units; TNL is the total nighttime light (TNL) in the administrative units (i.e., the sum of all pixel values of nighttime light data in the administrative unit); and a and b are the regression coefficient and intercept, respectively. The sample numbers of the eastern and central regions were 87 and 80, respectively, for a total of 167 samples.

## 3. Results

### 3.1. Regression of TNL and GRP

The GRP includes the gross regional product of the total city (including non-urban populations, GRPTC) and the gross regional product of districts within the city (only urban populations, GRPDC); it is expressed in units of 10,000 yuan. The linear regression results of the two indicators and TNL are shown in Table 3 and Figure 3. All *p* values were less than 0.001, meaning that the night lights and GRPTC and GRPDC showed a significantly positive correlation.

The results in Figure 3a,b show that in the eastern and central regions, the regression *R*^2^ value of the GRPTC and the TNL for the LJ1-01 data (0.843) was higher than that for the NPP/VIIRS data (0.830). The results in Figure 3c,d show that in the eastern and central regions, the regression *R*^2^ value of the GRPDC and TNL for the LJ1-01 data (0.745) was higher than that of the same using the NPP/VIIRS data (0.681). The regression results for the two parameters show that the correlation between the LJ1-01 data and the GRP was higher than the same for the NPP/VIIRS data. In other words, the LJ1-01 data have greater potential to model the GRP, which is mainly due to the higher spatial resolution of the LJ1-01 data and the more detailed reflection available for the commercial activities at night. The results in Table 3 show that the correlation between the two types of nighttime light data in the whole region (GRPTC) was higher than for just urban regions (GRPDC), except for LJ1-01 data in the central region, indicating that the nighttime light data can better reflect the gross regional product of the entire city.

### 3.2. Regression of TNL and Population

The annual average population includes the annual average population of the total city (AAPTC), which includes some non-urban residents, while the annual average population of districts with the city (AAPDC) includes only urban residents; these are expressed in units of 10,000 people. The linear regression results of the two indicators and TNL are shown in Table 4 and Figure 4. All *p* values were less than 0.001, meaning that the night lights and AAPTC and AAPDC showed a significantly positive correlation.

The results in Figure 4a,b show that in the eastern and central regions, the regression *R*^2^ value for the AAPTC and LJ1-01 data (0.237) was lower than the same using the NPP/VIIRS data (0.305), indicating that the correlation between NPP/VIIRS data and AAPTC was higher than the same for LJ1-01 data. The results in Figure 4c,d show that in the eastern and central regions, the regression *R*^2^ value of AAPDC and LJ1-01 data (0.678) was higher than the same using NPP/VIIRS data (0.638), indicating that the correlation between LJ1-01 data and AAPDC was higher than the same for NPP/VIIRS data.

The comparison of Figure 4a with Figure 4c and Figure 4b with Figure 4d shows that the regression *R*^2^ values of the two kinds of nighttime light data in districts within the city were much lower than those in the entire city, indicating a higher correlation between nighttime light data and AAPDC and a lower correlation between nighttime light data and AAPTC. In combination with the values listed in Table 5, the analysis of the statistical results for the AAPTC and AAPDC, with total populations of 790.26 million and 285.61 million, respectively, showed a large difference between the two. As AAPTC includes a large amount of non-urban residents, unlike AAPDC, the former undertakes fewer activities and thus generates less light. Therefore, nighttime light data mainly reflect the night light activity in the municipal districts.

The eastern region includes Beijing, Tianjin, and Shanghai—three municipalities directly under the central government—whose AAPTC and AAPDC are the same. Linear regression results for the eastern region of the AAPDC with *R*^2^ values for two types of nighttime light data (LJ1-01: 0.314, NPP: 0.362) were better than those (LJ1-01: 0.174, NPP: 0.317) for the central region. The results also verified the conclusion that nighttime light data were more correlated with the number of people in municipal districts.

### 3.3. Regression of TNL and Electricity Consumption

Electricity consumption of districts in the city includes annual electricity consumption (AEC), electricity consumption for industrial (ECI), and household electricity consumption for urban and rural residential areas (HECURR); this is expressed in units of 10,000 kwh. The linear regression results for the three indicators and TNL are shown in Table 6 and Figure 5. All *p* values were less than 0.001, meaning that night lights and AEC, ECI, and HECURR showed significantly positive correlations.

The results in Figure 5a,b show that in the eastern and central regions, the regression *R*^2^ value of the AEC and LJ1-01 data (0.765) was higher than that of the same using NPP/VIIRS data (0.723). The results in Figure 5c,d show that in the eastern and central regions, the regression *R*^2^ of the ECI and LJ1-01 data (0.651) was higher than that using NPP/VIIRS data (0.646). The results in Figure 5e,f show that in the eastern and central regions, the *R*^2^ value of HECURR and LJ1-01 data regression (0.718) was higher than that using NPP/VIIRS data (0.659). The results in Table 6 show that except for the linear regression *R*^2^ value of ECI and TNL of the LJ1-01 data in the eastern region (0.625), which is slightly lower than that of the NPP/VIIRS data (0.634), the degree of correlation between other LJ1-01 data and the electricity consumption in districts in the city was higher than the same using NPP/VIIRS data. In other words, the LJ1-01 data have greater potential to simulate the electricity consumption in municipal districts.

The correlation between the two types of nighttime light data in Table 6 and ECI in the eastern region (LJ1-01: 0.625, NPP: 0.634) was much higher than that in the central region (LJ1-01: 0.384, NPP: 0.328). The correlation degree of HECURR in the eastern region (LJ1-01: 0.685, NPP: 0.626) was lower than that in the central region (LJ1-01: 0.769, NPP: 0.702). This suggested that the overall level of industrial development in the eastern region was relatively high, and so the correlation between ECI and the nighttime light data was closer. The overall level of industrial development in the central region was low, and so HECURR accounts for a larger proportion of the electricity consumption, which is more closely related to the nighttime light data.

### 3.4. Regression of TNL and Area of Land Used

The area of land used for districts in the city includes the area of land used for urban construction (ALUC, km^2^), area of land used for living (ALL, km^2^), and area of the city used for paved road (ACPR, 10,000 m^2^). The linear regression results of the three indicators and TNL are shown in Table 7 and Figure 6. All *p* values were less than 0.001, which means the night lights and ALUC and ALL and ACPR showed significantly positive correlations.

The results in Figure 6a,b show that in the eastern and central regions, the linear regression *R*^2^ value of ALUC and TNL of the LJ1-01 data (0.745) was higher than that using the NPP/VIIRS data (0.683), indicating that the correlation between the LJ1-01 data and ALUC was higher than of the same using the NPP/VIIRS data. The results in Figure 6c,d show that in the eastern and central regions, the linear regression *R*^2^ value of ALL and the LJ1-01 data (0.748) was higher than that of the NPP/VIIRS data (0.690), indicating that the correlation between ALL and the LJ1-01 data was higher than the same using the NPP/VIIRS data. For the area of the city used for paved road, in the eastern region, the regression *R*^2^ value of ACPR and the TNL of the LJ1-01 data (0.551) was lower than that of the NPP/VIIRS data (0.582); in the central region, the regression *R*^2^ value of ACPR and the TNL of the LJ1-01 data (0.837) was higher than that of the NPP/VIIRS data (0.722). In summary, the correlation between the LJ1-01 data and the area of land used for districts in the city was mostly higher than that using the NPP/VIIRS data. In other words, the LJ1-01 data had greater potential to simulate the land use area in municipal districts.

Table 7 shows that the degree of correlation of the ACPR data in the central region and nighttime light (LJ1-01: 0.837, NPP: 0.722) was much higher compared with the eastern region (LJ1-01: 0.551, NPP: 0.582). The reason is that the overall level of economic development in the central region is lower than that of the eastern region, and the light from roads accounts for a larger proportion of the brightness of city lights, so the ACPR correlates with the nighttime light data more closely.

## 4. Conclusions

In this study, two types of nighttime light images and 10 types of socio-economic statistical data within 87 and 80 statistical units in the eastern and central regions of China, respectively, were analyzed by linear regression. According the results in Table A1 in Appendix A, the following conclusions were drawn:
(1)Comparing the two types of nighttime light data showed that the LJ1-01 nighttime light data offered more potential than the NPP/VIIRS nighttime light data to model socio-economic statistical data. This was mainly due to the higher spatial resolution of the LJ1-01 data, resulting in a more detailed characterization of commercial activities at night. Thus, the LJ1-01 data can be used as an effective tool to establish a model that offers socio-economic indicators.(2)There were significant positive correlations between the 10 types of socio-economic statistics and TNL, and the GRP showed the greatest potential for modeling. The *R*^2^ value of the GRPTC and LJ1-01 data and NPP/VIIRS data in the eastern and central regions were 0.843 and 0.830, respectively, indicating that nighttime light was most closely related to the GRP.(3)Due to the large differences between the population of the entire city and the urban districts within the city, a large number of non-urban population activities generated less light; therefore, the nighttime light data mainly reflected the districts within the city with more light-involving activities at night, while the correlation between the AAPDC and LJ1-01 data was higher than with NPP/VIIRS data.(4)The overall level of industrial development in the eastern region was relatively high, and so the ECI was more closely related to the nighttime light data. The industrial development level in the central region was generally low. Therefore, HECURR occupied a larger proportion of the electricity consumption and was more closely related to nighttime light data.(5)The correlation between ACPR and nighttime light data in the central region (LJ1-01: 0.837, NPP: 0.722) was much higher than that in the eastern region (LJ1-01: 0.551, NPP: 0.582). The overall economic development level of the central region was lower than that of the eastern region, and the light from roads accounted for a large proportion of the urban light brightness. Therefore, ACPR was more closely related to nighttime light data.(6)For 10 types of socio-economic parameters, the gross regional product of districts within cities experienced the biggest improvement in the correlation between the LJ1-01 and NPP/VIIRS data, improving by 0.052, 0.155, and 0.064 in the eastern, central, and both regions, respectively.

Due to the absence of socio-economic statistics in 2018, this study used the latest socio-economic statistics (2016) as well as the 2018 LJ1-01 and NPP/VIIRS data for night light image modeling. Despite the two-year gap between these data sets, the two types of nighttime light data exhibited good performance in the modeling of socio-economic statistics, with the LJ1-01 data showing more potential than the NPP/VIIRS data. It can be inferred that when LJ1-01 data are used to model the socio-economic statistics of the same year, the correlation should be stronger [3]. Given the high potential of LJ1-01 nighttime light data for modeling socio-economic parameters in China, it is reasonable to predict that these data would be applicable to modeling socio-economic parameters elsewhere in the world, but such considerations were beyond the scope of this study and require further research.

LJ1-01 nighttime light data is an emerging data source. All released data were collected in 2018, which is not sufficient for comprehensive evaluation. The Wuhan university team is working hard to produce more high-quality LJ1-01 nighttime light images, which can be analyzed in multiple time series in more fields in the future.

## Figures and Tables

**Figure 1 sensors-19-01465-f001:**
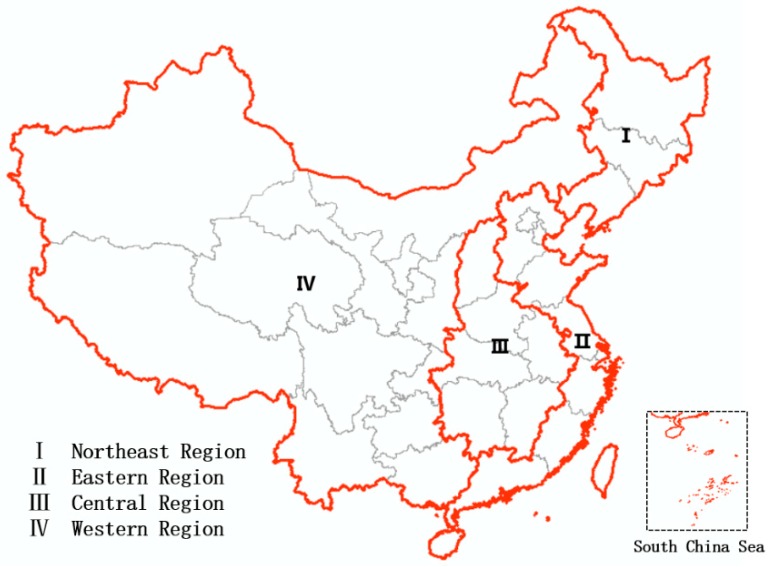
The four major Chinese economic regions in this study.

**Figure 2 sensors-19-01465-f002:**
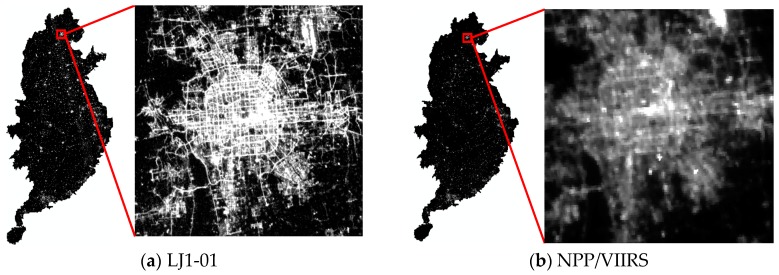
Nighttime light imagery of the study area. Red rectangle outlines the Beijing urban area. (**a**) LJ1-01 nighttime light imagery in 2018; (**b**) NPP/VIIRS nighttime light imagery in 2018.

**Figure 3 sensors-19-01465-f003:**
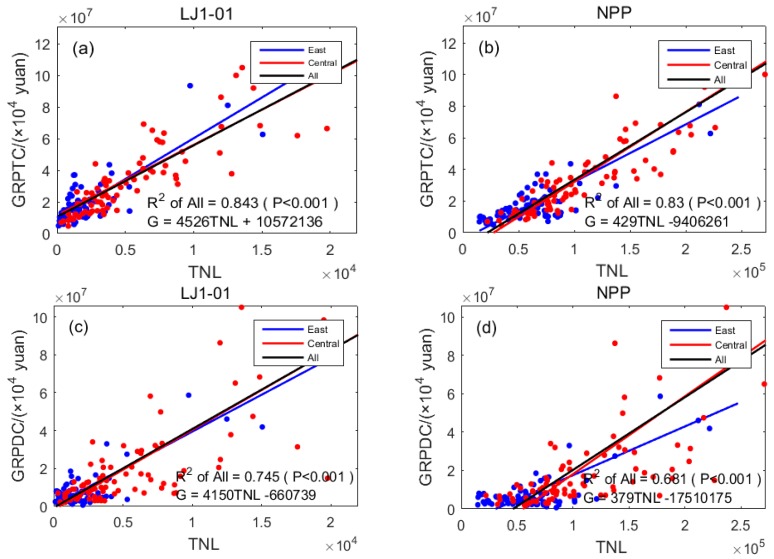
Scatter diagrams for linear regression results of GRP and TNL: (**a**) TNL of LJ1-01 data and GRPTC; (**b**) TNL of NPP/VIIRS data and GRPTC; (**c**) TNL of LJ1-01 data and GRPDC; (**d**) TNL of NPP/VIIRS data and GRPDC.

**Figure 4 sensors-19-01465-f004:**
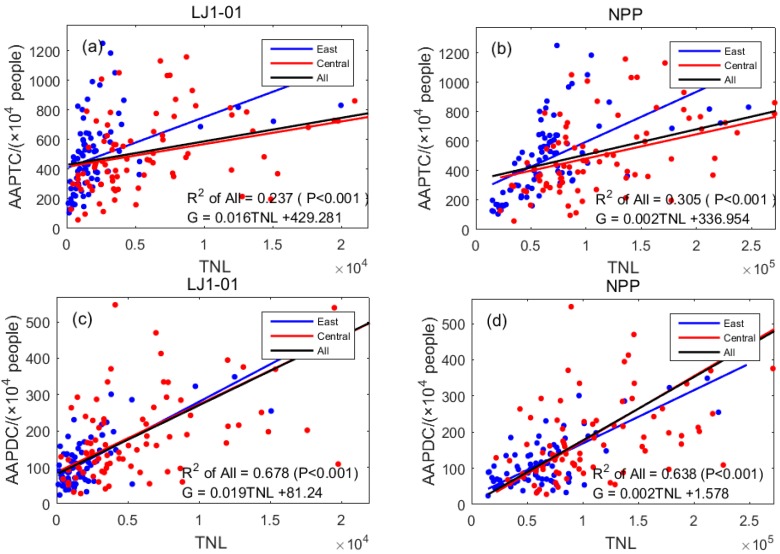
Scatter diagrams for linear regression results of population and TNL: (**a**) TNL of LJ1-01 data and AAPTC; (**b**) TNL of NPP/VIIRS data and AAPTC; (**c**) TNL of LJ1-01 data and AAPDC; (**d**) TNL of NPP/VIIRS data and AAPDC.

**Figure 5 sensors-19-01465-f005:**
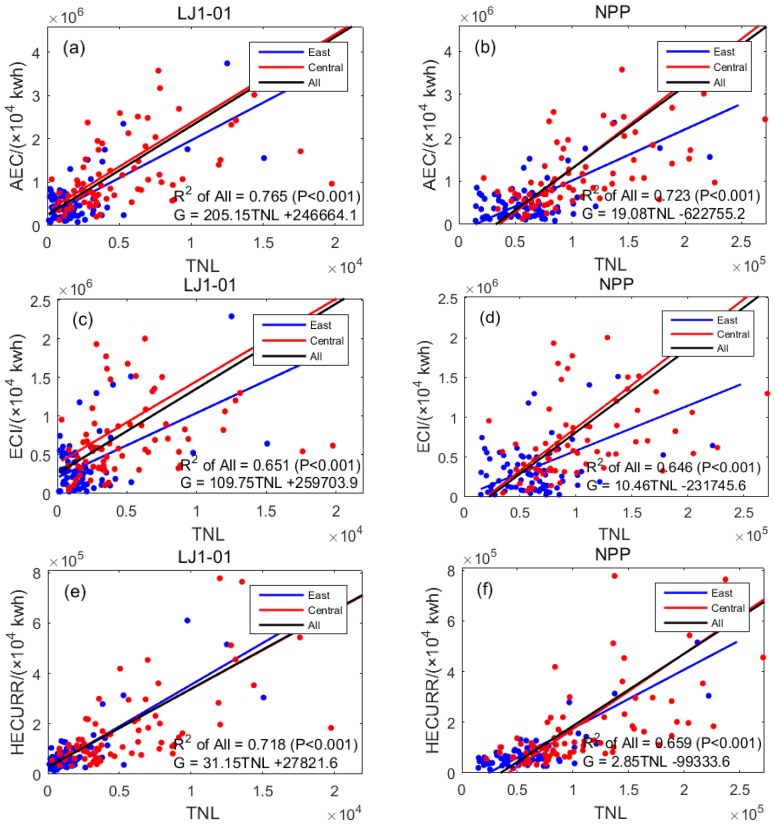
Scatter diagrams for linear regression results of electricity consumption and TNL: (**a**) TNL of LJ1-01 data and AEC; (**b**) TNL of NPP/VIIRS data and AEC; (**c**) TNL of LJ1-01 data and ECI; (**d**) TNL of NPP/VIIRS data and ECI; (**e**) TNL of LJ1-01 data and HECURR; (**f**) TNL of NPP/VIIRS data and HECURR.

**Figure 6 sensors-19-01465-f006:**
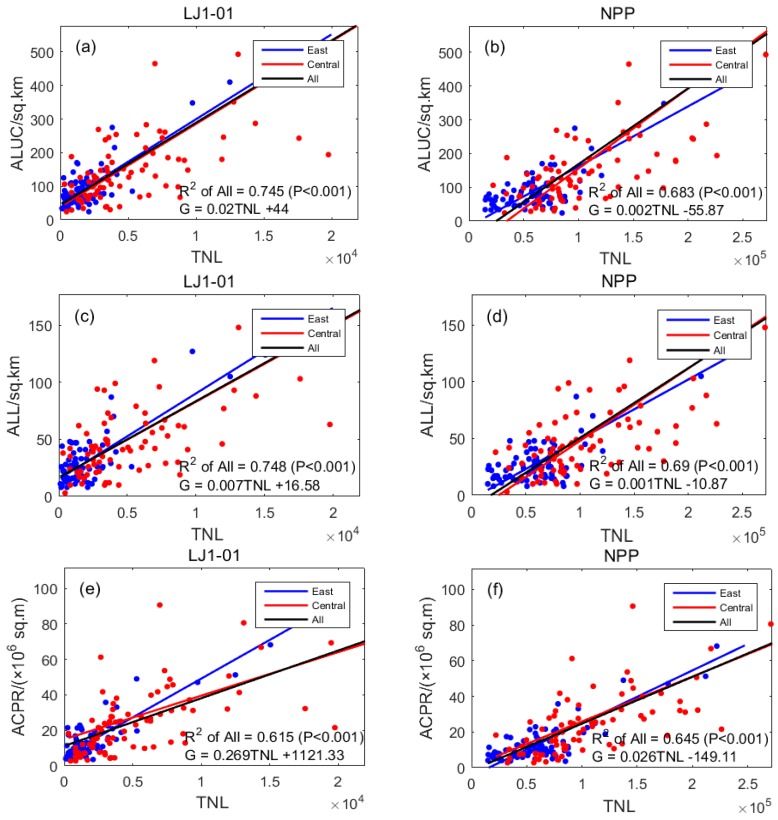
Scatter diagrams for the linear regression results of area of land in use and TNL: (**a**) the TNL of LJ1-01 data and ALUC; (**b**) the TNL of NPP/VIIRS data and ALUC; (**c**) the TNL of LJ1-01 data and ALL; (**d**) the TNL of NPP/VIIRS data and ALL; (**e**) the TNL of LJ1-01 data and ACPR; (**f**) the TNL of NPP/VIIRS data and ACPR.

**Table 1 sensors-19-01465-t001:** Specific administrative units studied.

Region	Municipality or Province	Number of Cities Counted	Sum
Eastern region	Beijing	1	87
	Tianjin	1	
	Hebei province	11	
	Shandong province	17	
	Shanghai	1	
	Jiangsu province	13	
	Zhejiang province	11	
	Fujian province	9	
	Guangdong province	21	
	Hainan province	2	
Central region	Shanxi province	11	80
	Anhui province	16	
	Jiangxi province	11	
	Henan province	17	
	Hubei province	12	
	Hunan province	13	
Sum	16	167	167

**Table 2 sensors-19-01465-t002:** Classification of statistical parameters.

Type of Statistics	Abbreviations	Name
Gross regional product	GRPTC	Gross regional product of total city (including non-urban)
	GRPDC	Gross regional product of urban districts within city
Population	AAPTC	Annual average population of total city (including non-urban)
	AAPDC	Annual average population of urban districts within city
Electricity consumption	AEC	Annual electricity consumption
	ECI	Electricity consumption for industries
	HECURR	Household electricity consumption in urban and rural residential areas
Area of land use	ALUC	Area of land use for urban construction
	ALL	Area of land use for living
	ACPR	Area of city paved road

**Table 3 sensors-19-01465-t003:** Linear regression *R*^2^ values for the GRP and TNL.

Region	East		Central		All	
Data	LJ1-01	NPP	LJ1-01	NPP	LJ1-01	NPP
GRPTC (Gross regional product of total city)	0.833	0.825	0.770	0.716	0.843	0.830
GRPDC (Gross regional product of districts within city)	0.721	0.669	0.771	0.616	0.745	0.681

**Table 4 sensors-19-01465-t004:** Linear regression *R*^2^ values for population and TNL.

Region	East		Central		All	
Data	LJ1-01	NPP	LJ1-01	NPP	LJ1-01	NPP
AAPTC (Annual average population of total city)	0.314	0.362	0.174	0.317	0.237	0.305
AAPDC (Annual average population of districts within city)	0.649	0.606	0.624	0.596	0.678	0.638

**Table 5 sensors-19-01465-t005:** Average annual population statistics of the study area (unit: 10,000 people).

Region	Municipality or Province	AAPTC (Annual Average Population of Total City)	AAPDC (Annual Average Population of Districts within City)
Eastern region	Beijing	1354	1354
	Tianjin	1036	1036
	Hebei province	1033	413
	Shandong province	9906	3439
	Shanghai	1446	1446
	Jiangsu province	7747	3569
	Zhejiang province	4893	1812
	Fujian province	3745	1085
	Guangdong province	8959	4609
	Hainan province	224	224
Central region	Shanxi province	3518	993
	Anhui province	6989	2085
	Jiangxi province	4792	1186
	Henan province	11,156	2290
	Hubei province	5305	1613
	Hunan province	6923	1407
Sum	-	79,026	28,561

**Table 6 sensors-19-01465-t006:** Linear regression *R*^2^ values for electricity consumption and TNL.

Region	East		Central		All	
Data	LJ1-01	NPP	LJ1-01	NPP	LJ1-01	NPP
AEC (Annual electricity consumption)	0.743	0.711	0.626	0.540	0.765	0.723
ECI (Electricity consumption for industries)	0.625	0.634	0.384	0.328	0.651	0.646
HECURR (Household electricity consumption in urban and rural residential areas)	0.685	0.626	0.769	0.702	0.718	0.659

**Table 7 sensors-19-01465-t007:** Linear regression *R*^2^ values for the area of land used and TNL.

Region	East		Central		All	
Data	LJ1-01	NPP	LJ1-01	NPP	LJ1-01	NPP
ALUC (Area of land use for urban construction)	0.717	0.657	0.804	0.726	0.745	0.683
ALL (Area of land use for living)	0.722	0.663	0.753	0.678	0.748	0.690
ACPR (Area of city paved road)	0.551	0.582	0.837	0.722	0.615	0.645

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
