# Peer review of "Evaluating the Potential of LJ1-01 Nighttime Light Data for Modeling Socio-Economic Parameters"

_sensors, 2019, doi:10.3390/s19061465_

Reviewer 1 Report

1) The text refers to the "The LJ1-01 satellite is the first professional nighttime light remote sensing satellite in the world".  This is untrue.  Both the DMSP and VIIRS low light imaging sensors were professionally designed - by engineers.  

2) The text should state that the low light imaging capability of the DMSP and VIIRS was designed to meet a requirement from meteorologists for nightly global visible imagery of clouds based on the reflectance of moonlight.  The detection of electric lighting was not a consideration in the design and was only discovered when the early images were viewed on the ground.

3) The text should state the design objectives of the instrument.

4) The text should reference the papers describing the Nightsat mission, proposed to NASA in 2011:  

Elvidge, C. D., Cinzano, P., Pettit, D. R., Arvesen, J., Sutton, P., Small, C., ... & Weeks, J. (2007). The Nightsat mission concept. International Journal of Remote Sensing, 28(12), 2645-2670.

Elvidge, C. D., Safran, J., Tuttle, B., Sutton, P., Cinzano, P., Pettit, D., ... & Small, C. (2007). Potential for global mapping of development via a nightsat mission. GeoJournal69(1-2), 45-53.

5) The text should reference the methods used to generate the VIIRS cloud-free composites used in the study: Elvidge, C. D., Baugh, K., Zhizhin, M., Hsu, F. C., & Ghosh, T. (2017). VIIRS night-time lights. International Journal of Remote Sensing38(21), 5860-5879.

6) The text should reference this paper, which details the improvements of the VIIRS relative to DMSP: Elvidge, C. D., Baugh, K. E., Zhizhin, M., & Hsu, F. C. (2013, April). Why VIIRS data are superior to DMSP for mapping nighttime lights. In Proceedings of the Asia-Pacific Advanced Network (Vol. 35, No. 62).

7) The text should note that the instrument has no long-wave infrared channel for use in discriminating clear versus cloudy areas.  How will the data be processed to exclude cloudy observations?

8) How is the instrument calibrated?  What is the radiometric accuracy of the calibrated data?

9) What is the collection plan?  Will the instrument collect data on every possible nighttime pass?  Or will it only collect on land areas?  

10) What is the data access plan for the nightly data?  Will there be an openly accessible archive that researchers can search and download from?  This is the case for VIIRS data.

11) What is the data access plan for comprehensive annual composites?  Will these be made available for open access download by all those who have an interest to work with the data?  This is the case for the VIIRS monthly and annual data products.

Author Response

Thank you for your valuable comments. Attachment please find the response.

Reviewer 2 Report

This is an interesting paper that reviews new advances in night-light remote sensing. In order for the information and results to be conveyed to the readers in the clearest possible way, I would suggest a few minor revisions at this stage.

First, not every reader will be familiar with the LJ1-01 satellite. Therefore, it would be ideal if a brief introduction to the satellite could be provided, perhaps on p. 2 line 46 where the date of launch is described. Here, the authors could consider providing answers to the following questions that I had, which might occur to other readers as well: Who launched the satellite (Wuhan University), why it was developed (university or government initiative), and with what purpose (i.e. just collecting night light imagery for scientific analysis, or are there other intended commercial or industrial applications, etc.)? What unit of radiance does LJ1-01 measure night lights in? In which file format is the data available, and can other researchers freely access the data? I tried accessing http://59.175.109.173:8888/parameters.html but the website does not appear to work.

Second, I am confused as to how the LJ1-01 data was processed. This information is provided in section 2.3 for the VIIRS data, but not LJ1-01. In general, the Data Preprocessing section could be made clearer. In which datasets were the negative pixels removed - the LJ1-01 data, VIIRS data, or the superimposed data? Some authors (e.g. Elvidge et al. 2017) have dealt with noise in VIIRS data by adding a constant of two rather than removing pixels so as to reduce data loss. What is the basis for the authors using their described noise reduction method instead?

Third, the study is only conducted in China but since the data apparently has global coverage (p. 2 line 49), some hypotheses about how the data would perform worldwide could be described in the conclusion.

Fourth, the conclusion mentions which socioeconomic variables are most closely correlated with LJ1-01, but it would also be helpful to know in brief which variables experienced the biggest improvement in correlation between LJ1-01 and VIIRS.

Fifth, I find the abbreviations hard to follow. They might make sense in the text, but the tables might be easier to quickly interpret if the abbreviations were actually spelled out.

Sixth, I wonder if the authors wish to consider including just one large table that includes all of the correlations together for easier cross-referencing and comparison rather than five separate tables.

Seventh, the meaning of "districts under city" is unclear. It might be better phrased as "districts within city." Even with this phrasing though, some of the meaning is still lost on me. For instance, the authors write on p. 6 lines 172-173, "In addition to AAPDC, AAPTC also includes a large number of non-urban population." How can the AAPTC include a large non-urban population if the measure supposedly refers to districts within a city? And what does "In addition to AAPDC" mean in this context?

Reference:

Elvidge, C. D., Hsu, F. C., et al. (2017) ‘Lighting Tracks Transition in Eastern Europe’, in Gutman, G. and Radeloff, V. (eds) Land-Cover and Land-Use Changes in Eastern Europe after the Collapse of the Soviet Union in 1991. New York: Springer, pp. 35–56.

Author Response

Thank you for your valuable comments. Attachment please find the response.

Round  2

Reviewer 1 Report

  The methods used to discriminate clear versus cloud impacted observations should be described.

Section 2.2 starts with "All LJ1-01 cloud-free nighttime light imagery data satellite data are provided to users free of charge [23]."  This is good.  But why only the cloud-free images?  Why not all of the nighttime observations? How are the cloud-free images selected?

The paper should refer to a new paper which finds that the VIIRS DNB has the lowest detection limits of all the VIIRS bands for fires and flares. Elvidge, Christopher D., Mikhail Zhizhin, Kimberly Baugh, Feng Chi Hsu, and Tilottama Ghosh. "Extending Nighttime Combustion Source Detection Limits with Short Wavelength VIIRS Data." Remote Sensing 11, no. 4 (2019): 395.

Author Response

(The authors gave the same response as above.)
